# Mineralocorticoid Receptor Antagonist (Potassium Canrenoate) Does Not Influence Outcome in the Treatment of COVID-19-Associated Pneumonia and Fibrosis—A Randomized Placebo Controlled Clinical Trial

**DOI:** 10.3390/ph15020200

**Published:** 2022-02-05

**Authors:** Katarzyna Kotfis, Igor Karolak, Kacper Lechowicz, Małgorzata Zegan-Barańska, Agnieszka Pikulska, Paulina Niedźwiedzka-Rystwej, Miłosz Kawa, Jerzy Sieńko, Aleksandra Szylińska, Magda Wiśniewska

**Affiliations:** 1Department of Anesthesiology, Intensive Therapy and Acute Intoxications, Pomeranian Medical University, 70-111 Szczecin, Poland; igor.karolak@gmail.com (I.K.); kacper.lechowicz@gmail.com (K.L.); mazegan@wp.pl (M.Z.-B.); apikulska@wp.pl (A.P.); 2Institute of Biology, University of Szczecin, 71-412 Szczecin, Poland; paulina.niedzwiedzka-rystwej@usz.edu.pl; 3Department of Radiology, Pomeranian Medical University, 70-111 Szczecin, Poland; kawamilosz@gmail.com; 4Department of General and Transplantation Surgery, Pomeranian Medical University, 70-111 Szczecin, Poland; jsien@poczta.onet.pl; 5Department of Medical Rehabilitation and Clinical Physiotherapy, Pomeranian Medical University, 71-111 Szczecin, Poland; aleksandra.szylinska@gmail.com; 6Clinical Department of Nephrology, Transplantology and Internal Medicine, Pomeranian Medical University, 70-111 Szczecin, Poland; mwisniewska35@gmail.com

**Keywords:** SARS-CoV-2, COVID-19, pulmonary fibrosis, pneumonia, acute respiratory distress syndrome, ARDS, mortality, potassium canrenoate

## Abstract

In December 2019 the SARS-CoV-2 virus appeared in the world, mainly presenting as an acute infection of the lower respiratory tract, namely pneumonia. Nearly 10% of all patients show significant pulmonary fibrotic changes after the infection. The aim of this study was to evaluate the effectiveness and safety of potassium canrenoate in the treatment of COVID-19-associated pneumonia and pulmonary fibrosis. We performed a randomized clinical trial (RCT) of potassium canrenoate vs placebo. A total of 55 patients were randomized and 49 were included in the final analysis (24 allocated to the intervention group and 25 allocated to the control group). Patients were assessed by physical examination, lung ultrasound, CT imaging and blood samples that underwent biochemical analysis. This RCT has shown that the administration of potassium canrenoate to patients with COVID-19 induced pneumonia was not associated with shorter mechanical ventilation time, shorter passive oxygenation, shorter length of hospitalization or less fibrotic changes on CT imaging. The overall mortality rate was not significantly different between the two groups. Adverse events recorded in this study were not significantly increased by the administration of potassium canrenoate. The negative outcome of the study may be associated with the relatively small number of patients included. Any possible benefits from the use of potassium canrenoate as an antifibrotic drug in COVID-19 patients require further investigation.

## 1. Introduction

In December 2019 SARS-CoV-2 virus appeared in the world, mainly presenting as an acute infection of the lower respiratory tract, namely pneumonia [1,2]. In March 2020, the World Health Organization (WHO) announced a pandemic in relation to the disease caused by the SARS-CoV-2 virus, known as Corona Virus Disease 2019 (COVID-19). The efforts of healthcare systems were to provide the most efficient and coordinated care and scientists around the world have focused on finding appropriate treatment for this specific type of pneumonia and its multiple long-term consequences [3]. COVID-19 has spread rapidly over a few months, all over the world, affecting patients of all ages. The majority of the cases are mild, viral illnesses, yet many patients suffer from severe respiratory diseases, i.e., pneumonia, respiratory failure, acute respiratory distress syndrome (ARDS) and non-respiratory symptoms, i.e., thrombosis and embolism, neuropathic pain, myositis, delirium, gastro-intestinal symptoms or long-term cognitive disturbances [4,5]. Initially, it was difficult to identify and study the long-term consequences of SARS-CoV-2 infection, but many, including the most common—fatigue, muscle weakness, sleep difficulties, anxiety, depression and pulmonary fibrosis—are now becoming apparent [6].

Part of the data comes from the observation of patients who recovered from SARS-CoV-1 infection, which may indicate what can be expected in the long-term from COVID-19 [7,8]. According to data from the World Health Organization (WHO), by 28 January 2022, the total number of COVID-19 cases worldwide was over 364,191,000 [9]. Calculations indicate that one-third of SARS-CoV-2 survivors may develop significant pulmonary fibrosis, reaching 80,000,000 people who may experience the chronic sequelae of pulmonary fibrosis. In addition, the COVID-19 pandemic is overwhelming many health systems, especially with the spread of the Omicron variant and the number of people infected with SARS-CoV-2 increasing every day. If the trend continues, the number of people at risk of post-infection chronic pulmonary fibrosis will continue to increase [10].

New data show a gradual recovery from the chronic effects of COVID-19, with approximately 30% having residual lung tissue damage 9 months after hospitalization, with nearly a third (or 10% of all patients) showing significant pulmonary fibrotic changes [11]. Further observation did not show any trends in reducing the severity or frequency of COVID sequelae [11].

More than 8000 cases and 700 deaths resulted from SARS-CoV-1 infection worldwide in the year 2003 [12]. In an observational study of 97 SARS-CoV-1 survivors, a one-year follow-up showed chest X-ray abnormalities in 28% of patients. The severity of changes in lung imaging was closely related to the degree of pulmonary functional impairment, and overall quality of life in survivors of SARS-CoV-1 infection was worse than in the age-matched comparison group [7]. Studies that followed SARS-CoV-1 survivors for 2 years and for 15 years showed similar results [8,13]. Based on the results of these studies, approximately one third of patients who survived SARS-CoV-1 infection had significant pulmonary fibrosis [8,13]. No specific cellular or molecular mechanisms leading to the development of pulmonary fibrosis in the course of SARS-CoV-2 infection, as in other viral diseases, have been identified. Some information is derived from previous studies on the SARS and MERS epidemic, as well as research on pulmonary fibrosis caused by other factors.

Numerous reports indicate that the use of mineralocorticoid receptor antagonists (MRAs) may be significant in fibrosis prevention [14,15,16]. Both spironolactone and potassium canrenoate, which is the only MRA clinically available for parenteral administration, traditionally used in treatment of hypertension and congestive heart failure, belong to this group [17]. Mineralocorticoid receptor (MR) pathway activation contributes to the pathophysiology of many diseases, being able to activate specific intracellular genomic and nongenomic pathways, leading to regulation of homeostasis of the cardiovascular system and becoming a crucial regulator of the growth and function of different cell types (fibroblasts, cardiomyocytes, vascular cells) [18,19]. Aldosterone is a physiological MR activator. It is partly responsible for the increase in extracellular matrix turnover, which is observed in pulmonary, cardiac and renal fibrosis and exerts its effect primarily on lung epithelium [14]. Studies show that a higher level of aldosterone may cause hypertension, alter inflammation and fibrosis and exacerbate cardiovascular disease [15]. The limitations of some of these studies are due to the use of animal models (rats or other rodents) or derived from molecular and plant studies as new substances are being investigated. Khalifa et al. in a review showed that there is a significant number of substances of natural origin with antiviral activity against human coronaviruses [20]. There is no confirmation of their activity against COVID-19 yet, therefore some of them are under clinical trials [20].

In preclinical animal studies spironolactone has been shown to act as an antioxidant and to protect organs from damage associated with oxidative stress by strengthening the antioxidative defense systems while inhibiting free radicals’ production [21]. Lung tissue treated with spironolactone showed a reduced number of cells such as lymphocytes, neutrophils, macrophages and eosinophils in the alveoli compared to those in which spironolactone was not used [22]. Lieber et al. showed that spironolactone treatment alleviates acute pneumonia caused not only by bleomycin but also by lipopolysaccharides [21]. In one preclinical study, Barut et al. analyzed the effect of spironolactone on lung damage due to intestinal ischemia and reperfusion [14]. The results suggest that initial treatment with spironolactone reduced neutrophil infiltration, nitric oxide synthase induction, oxidative stress and histopathological damage [23]. Similarly, Atalay et al. demonstrated the effectiveness of spironolactone in the treatment of acute lung damage [24], while Ji et al. indicated the therapeutic potential of spironolactone, which significantly reduces the inflammatory response of the lungs caused by bleomycin [22,25]. There are also no direct studies showing the beneficial effects of MRAs in postviral lung fibrosis or whether it could serve as a potential treatment for such a serious complication [26,27,28,29]. 

Therefore, the aim of this study was to evaluate the effectiveness of the intravenous form of mineralocorticoid receptor antagonist, potassium canrenoate, in the duration of invasive mechanical ventilation via endotracheal intubation or tracheotomy and duration of passive oxygen therapy in patients with COVID-19-associated pneumonia.

## 2. Results

### 2.1. Study Population

A total of 430 patients were assessed for eligibility in this study. After applying exclusion criteria 55 patients were randomized, and 49 were included in the final analysis (24 allocated to the intervention group and 25 allocated to the control group), Figure 1.

Baseline patient characteristics of both groups are shown in Table 1. Age, sex and body mass index was not significantly different between the placebo group and intervention group. Although there was a significantly higher rate of ischemic heart disease in the placebo group, rates of other preexisting comorbidities were not significantly different (Table 2). Similarly, the Clinical Frailty Scale score was higher in the placebo group and the difference was statistically important (3.76 vs. 3.17, *p* = 0.034).

There was no significant difference between the two groups in the chronic use of medications, with the exception that the intervention group had a significantly higher rate of oral antidiabetic drugs use (Table 3).

There were no significant differences between the groups in initial laboratory tests obtained on day 1 (Table 4) including: white blood cell counts, C-reactive protein, procalcitonin and blood ions. 

### 2.2. The Effect of Treatment on Survival, Mechanical Ventilation and Passive Oxygenation Time

The survival rate, length of mechanical ventilation, high flow nasal oxygenation therapy and passive oxygenation are shown in Table 5. The length of respiratory support of any kind was not significantly different between the groups. Furthermore, the length of hospital stay was also not significantly different between both groups (13.52 vs. 14.42 days, *p* = 0.617, Table 5).

The in-hospital mortality rate was 20.0% (5/25) in the control group versus 16.67% (4/24) in the intervention group. The difference between the two groups was not significant (*p* = 0.945). After 90 days the mortality rate among patients not lost to follow-up did not change between the groups (5/24 control vs. 4/22 intervention, *p* = 0.884), (Table 6).

### 2.3. The Effect of Treatment on Lung Imaging and Physical Performance 

Table 7 shows the sums of characteristic changes from all six planes assessed in TFS and the TFS scores in both groups. CT scans showed that after 90 days the percentages of fibrotic changes occurrence in lungs did not differ between the intervention and placebo groups at the time of examination and therefore the TFS did not differ between the groups as well (41.58 vs. 28.94, *p* = 0.513).

Figure 2 shows the progression of LUS scores in both groups acquired at different time points, the difference between the two groups was not statistically significant at any given timepoint.

Figure 3 shows the progression NEWS test scores in both groups acquired at different time points, the difference between the two groups was not statistically significant at any given timepoint.

Figure 4 shows the percent of predicted 6-min walk distance in both groups at two follow-up time points. The distances achieved during 6MWT did not differ significantly between the groups on both the 30th (*p* = 0.665) and the 90th day (*p* = 0.519).

### 2.4. Adverse Events

The incidence of adverse events in the two groups is summarized in Table 6. There were no significant differences in the incidence of adverse events between the two groups. The incidence of pneumothorax was slightly higher in the intervention group, but the difference was not statistically significant (*p* = 0.219). The incidence of hyperkalemia was also more frequent in the intervention group, but not statistically significant (*p* = 0.281). 

There were also no significant differences between the groups regarding the laboratory tests obtained on day 7 (Table 8) including: white blood cell counts, C-reactive protein, procalcitonin and blood ions.

## 3. Discussion

As the SARS-CoV-2 infections continue to spread, a need arises to implement a prophylaxis of COVID-19 sequelae, including lung fibrosis. In the present study, we investigated the efficacy and safety of potassium canrenoate in severe COVID-19 patients as a potential agent for prevention of COVID-19 associated pulmonary fibrosis. 

Various studies have suggested using MRA as an antifibrotic treatment for viral infections, especially coronavirus [10]. It is worth noting that spironolactone is the preferred inhibitor of the renin–angiotensin–aldosterone (RAA) system in COVID-19 [30]. The present study is the first clinical trial to investigate the effect of potassium canrenoate on the prevention and treatment of pulmonary fibrosis due to COVID-19 pneumonia. In addition to our research, other clinical trials of MRAs are underway, but the results have not yet been published (ClinicalTrials.gov (accessed on 10 January 2022) Identifiers: NCT04643691, NCT04826822, NCT04345887).

### 3.1. Effect of Potassium Canrenoate on the Lung Fibrosis Process 

Our study did not show any significant benefit from implementation of additional treatment with potassium canrenoate in patients with moderate to severe COVID-19 pneumonia. This might result from many reasons. 

It is a relatively small randomized clinical trial and the number of patients who reported for the final follow-up visit and underwent CT examination on day 90 was therefore not numerous. Although there are some beneficial tendencies in the intervention group, none are statistically significant and the antifibrotic effect of potassium canrenoate requires further large-scale research.

Another matter is the patient selection and disease severity. Pulmonary fibrosis is more likely to occur in patients with severe clinical conditions, especially in patients with high levels of inflammatory markers [31]. The significant increase in the production of many cytokines and growth factors may lead to impaired healing and excessive scarring [32,33,34]. There are reports on the beneficial effect of MRAs on endothelial inflammation in SARS-CoV-2 infection [35]. Some studies indicate there is a therapeutic potential of spironolactone, which significantly reduces the inflammatory response of the lungs after injury [22,25]. Taking into consideration the direct effects on the pulmonary endothelium and the indirect effects on inflammatory response and cytokine production, potassium canrenoate may be of beneficial effect for patients with already developed severe COVID-19 pneumonia. A study by Umemura et al. suggests an antifibrotic potential of nintedanib in ICU patients [36]. That might suggest a need to implement further studies including ICU patients. Despite a higher risk of fibrosis in ICU patients, fibrosis has also been documented in patients who did not require mechanical ventilation [37,38]. 

The ideal timing and duration of intervention with potassium canrenoate during COVID-19 pneumonia remain unknown. As one of the inclusion criteria was the blood oxygen saturation level under 94% indicating at least a partial destruction of lung tissue, the timing of intervention might have been belated. There are some concerns that premature immunomodulation may inhibit host antiviral immunity and delay viral clearance, while delaying immunomodulation may prove futile if acute pulmonary injury is advanced [39]. However, some studies suggest that chronic use of MRA might be associated with lower COVID-19 infection probability, though with no difference in complications due to COVID-19 [40].

### 3.2. The Effect of Potassium Canrenoate on Mortality 

Despite the suggested effects of potassium canrenoate on lung fibrosis, the overall mortality rate was not significantly different between the two groups. Some explanations for these conflicting results might be attributed to the speculation that attenuating pulmonary fibrosis with potassium canrenoate might not contribute to reducing acute-phase deaths. Pulmonary fibrosis is a late pathological finding associated with late death. A study by Thille et al. shows that of 159 autopsies of patients with ARDS, pulmonary fibrosis had developed in only 4% of the patients with a disease duration of less than 1 week, 24% of those with a disease duration of 1 to 3 weeks and 61% of those with a disease duration of greater than 3 weeks [41]. Additionally, it was observed that the proportion of patients treated with placebo who had ischemic heart disease, a comorbidity associated with poorer outcomes in COVID-19, was significantly higher than in the intervention group.

### 3.3. Safety of Potassium Canrenoate 

Adverse events recorded in this study were not significantly increased by the administration of potassium canrenoate. Although, some side effects of potassium canrenoate were reported to include hyperkalemia, hyponatremia, and hypovolemia. Blood ion levels did not differ significantly between the groups, however there might be a tendency for the intervention group to develop hyperkalemia, which is the most common side effect of potassium canrenoate. 

### 3.4. Limitations

First, although randomized, it was a single center study, and this might have generated bias. Second, the number of patients included in this study was relatively small. Further large-scale randomized trials are needed to thoroughly evaluate the effects of potassium canrenoate on the treatment of patients with COVID-19. Moreover, as the intervention may trend with an improved 6MWD, it may be argued that the difference in trial design achieved by including more patients, powering the study for improvement in 6MWD or adjusting the doses of the investigational drug would have shown clinical and statistical significance. 

## 4. Materials and Methods 

### 4.1. Ethics

This prospective phase IV randomized clinical trial (RCT) was performed between December 2020 and August 2021 in a University Hospital no. 2 of the Pomeranian Medical University in Szczecin, Poland. The study received approval of the Ethics Committee Board at the Pomeranian Medical University in Szczecin, Poland (ICE consent, no. 0012/100/2020, date 29 June 2020) and was registered at ClinicalTrials.gov (accessed on 10 January 2022) (identifier NCT04912011). 

### 4.2. Study Population

Patients of both genders, between the age of 18 and 90 years were included in the study, after being provided with detailed information regarding the study and signing an informed consent form (ICF). The research was conducted according to the Declaration of Helsinki.

### 4.3. Inclusion Criteria

Patients of both sexes, 18–90 years of age.Patient requiring oxygen therapy, blood oxygen saturation level <94%.Confirmed COVID-19 infection (rt-PCR).At least one risk factor for increased mortality during COVID-19 currently published in the literature e.g., smoking, hypertension, diabetes, cardiovascular disease.Documented informed consent according to ICH-GCP and national regulations.

### 4.4. Exclusion Criteria

Chronic bronchitis, emphysema, interstitial lung disease or other history of lung disease.Contraindications to the use of spironolactone.Hypersensitivity to spironolactone or any of the excipients.Pregnant patients (pregnancy test will be performed in every patient of reproductive age) and during lactation.Patients with mental illness or dementia who are unable to give informed consent to the examination.ARDS caused by another viral infection (SARS-CoV-2 negative).ARDS from other causes/trauma.Ionic disorders: hyperkalemia, hyponatremia.Adrenal crisis.Acute and chronic renal failure, creatinine clearance less than 30 mL/min.Anuria.Porphyria.Chronic use of MRA drugs from spironolactone group.

### 4.5. Clinical Experiment Measures

Consecutive patients were randomized using a computer-generated list of numbers to participate either in the experiment arm (Intervention group) who received 200 mg of potassium Canrenoate potassium (Aldactone) dissolved in 100 mL of 0.9% sodium chloride intravenously twice a day for 7 days or to the control arm (Placebo group) who received 100 mL of 0.9% sodium chloride intravenously twice a day for 7 days.

### 4.6. Outcome Measures

#### 4.6.1. Primary Outcome Measures

Duration of invasive mechanical ventilation via endotracheal intubation or tracheotomy (observation time 30 days).Duration of passive oxygen therapy (Observation time 30 days).

#### 4.6.2. Secondary Outcome Measures

Intensive Care Unit length of stay (LOS) (time frame 30 days).Total hospital length of stay (LOS) (time frame 90 days).Assessment of the dynamics of recovery of changes in lung ultrasound at 7 days.Assessment of the dynamics of recovery of changes in lung ultrasound at 30 days.Assessment of the dynamics of recovery of changes in chest computed tomography (CT) at 3 months (90 days).Assessment of mortality at 30 days.Assessment of mortality at 90 days.Six-minute walk test (6MWT) at 30 days.Six-minute walk test (6MWT) at 90 days.

### 4.7. Lung Ultrasound (LUS), Lung CT Evaluation and 6-min Walk Test

The LUS protocol to assess dynamics of regression of changes in lung ultrasound was developed for this study and was based on the available literature. The intensity of changes was assessed on a 4-point scale, assigning each area a score from 0 to 3 according to the following criteria [42].

Score 0: The pleural line is continuous and regular. There are horizontal artifacts, the so-called A lines. There are no more than two B lines.

Score 1: The pleural line is irregular, jagged. Vertical areas of white are shown below. There are more than two B lines.

Score 2: The pleural line is broken. Below the defect, darker areas of various sizes (consolidations) and associated white areas below the consolidated area (C lines) appear.

Score 3: Study area shows a dense and multi-regionally white lung with or without major consolidations. An air bronchogram may appear.

Each lung was examined in six sectors (two anterior, two lateral and two posterior). A given sector was assigned the highest score according to the images shown in the area [43].

The dynamics of resolution of changes in chest CT scans was assessed based on the Total Fibrosis Score [44]. The assessment was made by one radiologist to avoid discrepancies in the interpretation of CT images.

The 6-min walk tests (6MWT) were also performed by one investigator to avoid discrepancies in the test results. The equations used to calculate the 6-min walk test predicted distance were as follows:

Predicted distance for men = (7.57 ∗ height) − (5.02 ∗ age) − (1.76 ∗ weight) − 309

Predicted distance for women = (2.11 ∗ height) − (2.29 ∗ weight) − (5.78 ∗ age) + 667

### 4.8. Statistical Analysis

The sample size was calculated to demonstrate statistical significance of differences in the assessment of the duration of invasive mechanical ventilation via endotracheal intubation or tracheotomy (hours) at 48 h after admission, assuming the standard significance level of the test *p* = 0.05 and power of 0.90. Additionally, it was assumed that the standard deviation (SD) of the length of the duration of invasive mechanical ventilation via endotracheal intubation or tracheotomy time would be 48 h, and that when assessing statistical significance, the Student’s *t*-test for independent samples was used. With the above information taken into the assessment by the statistician the study size was calculated to include 23 patients per arm, with a minimum total of 46 patients. The research project assumed the number of patients in each group should be at the level of 25, because this number was found to be achievable with the incurred costs, study time and availability of patients with predetermined inclusion and exclusion criteria. The participants were randomly divided into one of the two groups (according to the randomization table generated from the www.randomiser.com (accessed on 28 August 2020).

## 5. Conclusions

This randomized placebo-controlled study has shown that the administration of potassium canrenoate to patients with COVID-19 induced pneumonia was not associated with shorter mechanical ventilation time, shorter passive oxygenation, shorter length of hospitalization or less fibrotic changes on CT imaging. The overall mortality rate was not significantly different between the two groups. Adverse events recorded in this study were not significantly increased by the administration of potassium canrenoate. The negative outcome of the study may be associated with the relatively small number of patients included. Any possible benefits from the use of potassium canrenoate as an antifibrotic drug in COVID-19 patients require further investigation.

## Figures and Tables

**Figure 1 pharmaceuticals-15-00200-f001:**
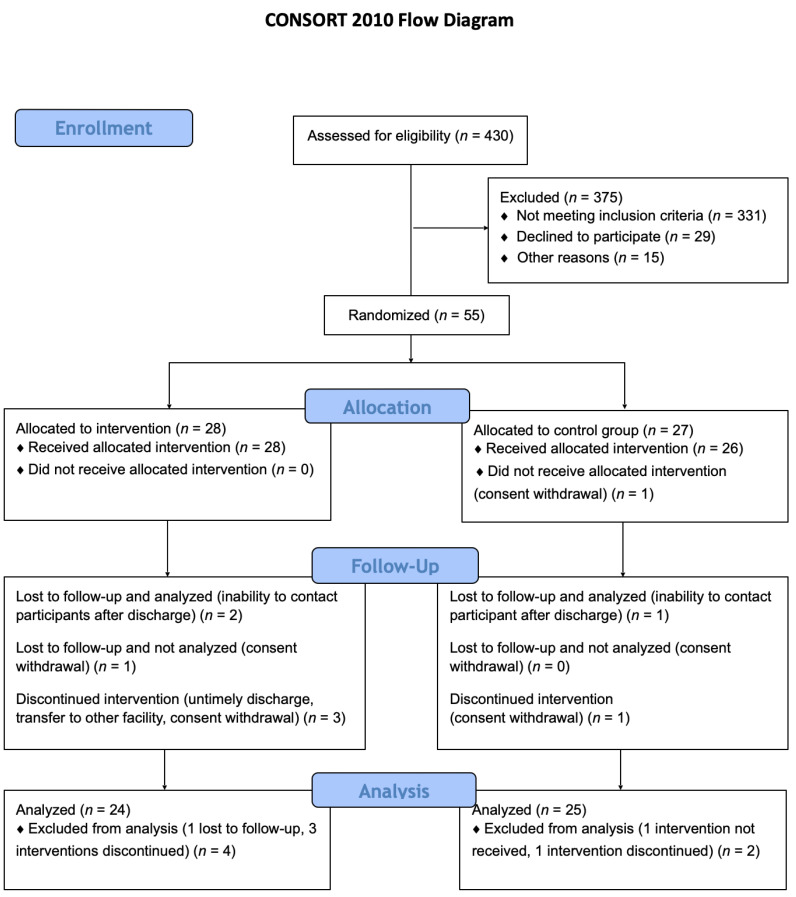
Study flow-chart. A total of 430 patients were assessed for eligibility in this study. After applying exclusion criteria 55 patients were randomized and 49 were included in the final analysis (24 allocated to the intervention group and 25 allocated to the control group).

**Figure 2 pharmaceuticals-15-00200-f002:**
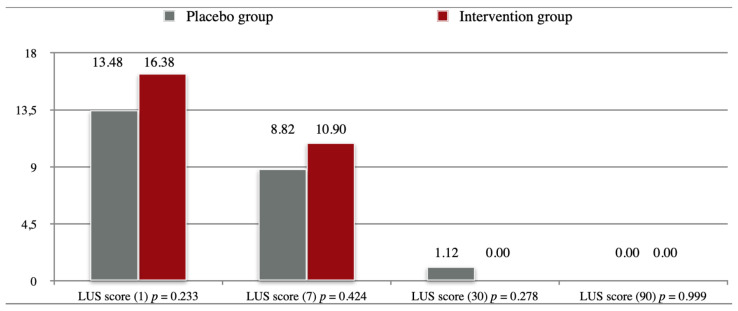
Lung Ultrasound Scores. Legend: Day of examination in brackets.

**Figure 3 pharmaceuticals-15-00200-f003:**
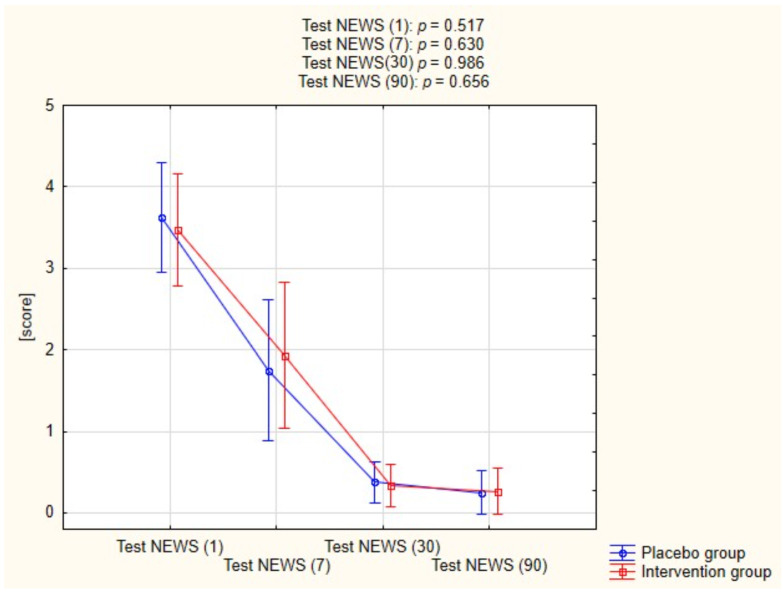
Assessment of Test NEWS in both groups.

**Figure 4 pharmaceuticals-15-00200-f004:**
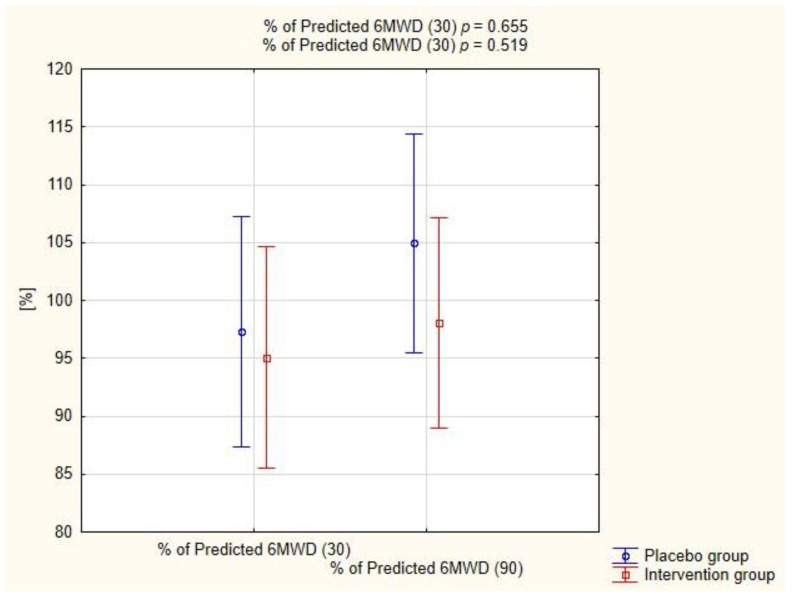
Assessment of predicted 6-min walk distance (6MWD) in both groups.

**Table 1 pharmaceuticals-15-00200-t001:** Baseline characteristics.

Variables		Placebo Group (*n* = 25)	Intervention Group (*n* = 24)	*p*-Value
Age [years], mean ± SD; Me		63.84 ± 14.75; 66.00	61.54 ± 9.06; 64.00	0.513
Gender [male], *n* (%)		16 (64.00)	10 (41.67)	0.200
BMI [kg/m^2^], mean ± SD; Me		30.57 ± 4.63; 29.05	30.92 ± 4.10; 30.78	0.780
Smoking, *n* (%)	No	12 (48.00)	14 (58.33)	0.204
	Yes	3 (12.00)	0 (0.00)	
	Quit >1 month	10 (40.00)	10 (41.67)	
Alcohol use, *n* (%)	No	7 (29.17)	9 (37.50)	0.734
	Yes	2 (8.33)	1 (4.17)	
	Occasionally	15 (62.50)	14 (58.33)	
CFS [1,2,3,4,5,6,7], (mean ± SD; Me)		3.76 ± 1.01; 4.00	3.17 ± 0.70; 3.00	0.034

Legend: BMI—body mass index, CFS—clinical frailty scale, Me—median, SD—standard deviation, *p*—statistical significance.

**Table 2 pharmaceuticals-15-00200-t002:** Comorbidities of patients included in the study.

Co-Morbidities	Placebo Group (*n* = 25)	Intervention Group (*n* = 24)	*p*-Value
Arterial hypertension, *n* (%)	16 (64.00)	15 (62.50)	0.851
Ischemic heart disease, *n* (%)	7 (28.00)	0 (0.00)	0.017
Myocardial infarction, *n* (%)	4 (16.00)	0 (0.00)	0.128
Chronic heart failure, *n* (%)	3 (12.00)	0 (0.00)	0.248
Atrial Fibrillation, *n* (%)	4 (16.00)	0 (0.00)	0.128
Hypercholesterolemia, *n* (%)	8 (32.00)	3 (12.50)	0.196
TIA, *n* (%)	1 (4.00)	0 (0.00)	0.984
Diabetes, *n* (%)	4 (16.00)	10 (41.67)	0.095
Peripheral vascular disease, *n* (%)	5 (20.00)	1 (4.17)	0.209
Peptic ulcer disease, *n* (%)	1 (4.00)	0 (0.00)	0.984
Thyroid disease, *n* (%)	4 (16.00)	4 (16.67)	0.746
Active NPL, *n* (%)	2 (8.00)	2 (8.33)	0.632
Depression, *n* (%)	1 (4.00)	0 (0.00)	0.984

Legend: NPL—neoplasm; TIA—transient ischemic attack, *p*—statistical significance.

**Table 3 pharmaceuticals-15-00200-t003:** Medications taken by the patient before admission.

Medications	Placebo Group (*n* = 25)	Intervention Group (*n* = 24)	*p*-Value
Aspirin, *n* (%)	7 (29.17)	1 (4.17)	0.053
ADP Inhibitors, *n* (%)	2 (8.33)	0 (0.00)	0.470
NOAC, *n* (%)	1 (4.17)	0 (0.00)	0.984
Beta-blockers, *n* (%)	10 (40.00)	6 (25.00)	0.415
ACE-I/Sartans, *n* (%)	12 (50.00)	11 (45.83)	0.999
Ca-blockers, *n* (%)	6 (25.00)	3 (12.50)	0.459
Statins, *n* (%)	6 (25.00)	2 (8.33)	0.245
Nitrates, *n* (%)	1 (4.17)	0 (0.00)	0.999
Diuretics, *n* (%)	7 (28.00)	8 (33.33)	0.924
Bronchodilators, *n* (%)	1 (4.00)	0 (0.00)	0.984
Oral hypoglycemic drugs, *n* (%)	2 (8.00)	10 (41.67)	0.016
Insulin, *n* (%)	1 (4.00)	2 (8.33)	0.971
Levothyroxine, *n* (%)	4 (16.00)	3 (12.50)	0.953
Opioids, *n* (%)	0 (0.00)	1 (4.17)	0.984

Legend: ADP inhibitors—adenosine diphosphate receptor inhibitors, NOAC—new oral anticoagulants, ACE-I—angiotensin-converting enzyme inhibitor, n—number of patients, NSAIDs—nonsteroidal anti-inflammatory drugs, *p*—statistical significance.

**Table 4 pharmaceuticals-15-00200-t004:** Laboratory results on Day 1.

Variables	Placebo Group (*n* = 25)	Intervention Group (*n* = 24)	*p*
	Mean ± SD; Me	Mean ± SD; Me	
White blood cells [G/L]	7.52 ± 3.06; 7.59	8.19 ± 3.00; 8.41	0.440
Neutrophils [G/L]	6.02 ± 2.84; 5.74	6.64 ± 2.98; 6.69	0.461
Lymphocytes [G/L]	0.95 ± 0.32; 0.90	1.05 ± 0.30; 0.98	0.266
Red blood cells [T/L]	4.20 ± 0.63; 4.27	4.30 ± 0.41; 4.20	0.523
Platelets [G/L]	260.08 ± 93.54; 245,00	317.04 ± 132.56; 265.00	0.091
Hemoglobin [mmol/L]	7.90 ± 0.99; 7.90	7.97 ± 1.11; 7.90	0.836
Hematocrit [l/l]	0.37 ± 0.05; 0.38	0.38 ± 0.04; 0.37	0.574
C-reactive protein [mg/dL]	71.08 ± 44.78; 76.04	95.58 ± 65.37; 80.14	0.135
Interleukin-6 [pg/mL]	46.68 ± 56.79; 24.90	64.97 ± 72.52; 41.00	0.332
Procalcitonin [ng/mL]	0.15 ± 0.12; 0.12	0.23 ± 0.35; 0.09	0.327
AST [U/L]	67.72 ± 89.11; 48.00	48.29 ± 20.78; 44.0	0.298
ALT [U/L]	52.88 ± 46.5; 35.00	48.79 ± 36.9; 38.0	0.734
LDH [U/L]	460.61 ± 174.88; 403.00	460.39 ± 154.07; 472.00	0.996
D-Dimer [ng/mL]	1799.32 ± 1902.33; 1158.00	2329.58 ± 2695.07; 1016.00	0.432
Ferritin [µg/L]	1262.88 ± 866.89; 983.50	948.83 ± 570.37; 835.00	0.223
K+ [mmol/L]	4.07 ± 0.54; 4.10	4.05 ± 0.51; 4.20	0.915
Na+ [mmol/L]	139.96 ± 3.32; 141.00	139.38 ± 3.88; 139.00	0.574
Cl- [mmol/L]	102.44 ± 3.96; 102.00	100.17 ± 4.73; 101.00	0.075

Legend: AST—aspartate transaminase, ALT—alanine transaminase, LDH—lactate dehydrogenase, Me—median, SD—standard deviation, *p*—statistical significance.

**Table 5 pharmaceuticals-15-00200-t005:** Data regarding treatment during hospitalization.

Variables	Placebo Group	Intervention Group	*p*
	Mean ± SD; Me	Mean ± SD; Me	
Length of hospital stay [days]	13.52 ± 5.84; 11.00	14.42 ± 6.57; 12.00	0.617
Length of ICU stay [h]	166.07 ± 88.89; 139.00	238.67 ± 217.01; 189.00	0.471
Passive oxygenation [days]	7.76 ± 4.48; 7.00	7.08 ± 5.61; 6.00	0.644
HFNOT [h]	90.13 ± 60.14; 88.00	112.31 ± 92.46; 88.50	0.580
Mechanical ventilation [h]	102.00 ± 59.06; 95.00	270.20 ± 224.39; 238.00	0.171
PO + HFNOT [days]	8.96 ± 4.54; 8.00	8.64 ± 6.90; 6.50	0.850
PO + HFNOT + MV [days]	10.15 ± 5.77; 8.00	10.99 ± 8.02; 8.50	0.678

Legend: ICU—Intensive Care Unit, PO—passive oxygenation, HFNOT—high-flow nasal oxygen therapy, MV—mechanical ventilation, Me—median, SD—standard deviation, *p*—statistical significance.

**Table 6 pharmaceuticals-15-00200-t006:** Complications and follow-up.

Variables	Placebo Group	Intervention Group	*p*
	*n* (%)	*n* (%)	
ICU admission	7 (28.00%)	6 (25.00%)	0.932
Death in hospital	5 (20.00%)	4 (16.67%)	0.945
Death after 90 days	5 (20.83%)	4 (18.18%)	0.884
Secondary infection	7 (28.00%)	5 (20.83%)	0.802
Pneumothorax	0 (0.00%)	3 (12.50%)	0.219
Hypotension (SBP < 100 mmHg)	5 (20.00%)	8 (33.33%)	0.463
Thromboembolic events	2 (8.00%)	3 (12.50%)	0.962
Hyperkalemia	4 (16.00%)	8 (33.33%)	0.281
Hypernatremia	2 (8.00%)	1 (4.17%)	0.971
Hypokalemia	3 (12.00%)	3 (12.50%)	0.702
Hyponatremia	1 (4.00%)	2 (8.33%)	0.971

Legend: ICU—Intensive Care Unit, Me—median, SD—standard deviation, *p*—statistical significance.

**Table 7 pharmaceuticals-15-00200-t007:** CT imaging results on day 90.

Variables	Placebo Group	Intervention Group	*p*
	Mean ± SD; Me	Mean ± SD; Me	
Total Honeycombing	7.11 ± 20.50; 0.00	0.00 ± 0.00; 0.00	0.148
Total Reticulation	26.53 ± 38.30; 15.00	26.94 ± 34.42; 7.00	0.972
Total Traction Bronchiectasis	5.32 ± 10.92; 0.00	2.00 ± 3.82; 0.00	0.226
Total Ground Glass Opacification	2.63 ± 11.47; 0.00	0.00 ± 0.00; 0.00	0.331
TFS	41.58 ± 74.07; 15.00	28.94 ± 36.39; 8.50	0.513

Legend: TFS—Total Fibrosis Score, Me—median, SD—standard deviation, *p*—statistical significance.

**Table 8 pharmaceuticals-15-00200-t008:** Laboratory results on Day 7.

Variables	Placebo Group	Intervention Group	*p*
	Mean ± SD; Me	Mean ± SD; Me	
White blood cells [G/L]	10.88 ± 5.77; 10.09	9.90 ± 4.19; 9.27	0.512
Neutrophils [G/L]	8.36 ± 5.81; 6.12	7.49 ± 4.41; 6.93	0.574
Lymphocytes [G/L]	1.57 ± 0.72; 1.44	1.60 ± 0.62; 1.78	0.875
Red blood cells [T/L]	4.31 ± 0.55; 4.33	4.26 ± 0.43; 4.19	0.714
Platelets [G/L]	372.70 ± 119.06; 365.00	385.87 ± 112.78; 380.00	0.702
Hemoglobin [mmol/L]	8.08 ± 0.77; 8.10	7.89 ± 0.97; 7.80	0.463
Hematocrit [l/l]	0.39 ± 0.04; 0.39	0.38 ± 0.04; 0.38	0.815
C-reactive protein [mg/dL]	25.71 ± 41.52; 7.60	28.35 ± 45.98; 10.50	0.841
Interleukin-6 [pg/mL]	60.56 ± 152.47; 11.00	24.20 ± 69.38; 5.30	0.317
Procalcitonin [ng/mL]	5.43 ± 23.14; 0.07	0.20 ± 0.49; 0.06	0.337
AST [U/L]	37.61 ± 29.22; 29.00	32.35 ± 14.63; 29.00	0.452
ALT [U/L]	65.22 ± 32.54; 67.00	59.05 ± 38.78; 49.00	0.579
LDH [U/L]	340.44 ± 164.33; 269.00	313.95 ± 100.17; 301.00	0.579
D-Dimer [ng/mL]	1719.48 ± 1826.09; 1105.00	1782.45 ± 1607.25; 1312.00	0.903
Ferritin [µg/L]	1495.19 ± 2205.69; 857.00	923.11 ± 816.19; 678.50	0.343
K+ [mmol/L]	4.45 ± 0.39; 4.50	4.66 ± 0.65; 4.65	0.198
Na+ [mmol/L]	139.22 ± 5.33; 138.00	139.41 ± 3.32; 139.50	0.885
Cl- [mmol/L]	102.05 ± 4.39; 100.50	100.95 ± 2.76; 101.50	0.350

Legend: AST—aspartate transaminase, ALT—alanine transaminase, LDH—lactate dehydrogenase, Me—median, SD—standard deviation, *p*—statistical significance.

## Data Availability

The dataset is not available in a public database due to legal reasons. Anonymous data may be provided by the corresponding author upon a reasonable request from a researcher.

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
