# Peer review of "Mineralocorticoid Receptor Antagonist (Potassium Canrenoate) Does Not Influence Outcome in the Treatment of COVID-19-Associated Pneumonia and Fibrosis—A Randomized Placebo Controlled Clinical Trial"

_pharmaceuticals, 2022, doi:10.3390/ph15020200_

Round 1

Reviewer 1 Report

Thank you for allowing me to review this manuscript and your clinical trial results evaluating the use of potassium canrenoate in the treatment/prevention of pneumonia and pulmonary fibrosis. 

Overall, while the trial has a general negative result, the presentation of the data is sound and appropriate conclusions have been drawn. It is in this reviewer's opinion that negative results in clinical trials are still results, and more trials of this type should be published. There is only a few minor commentaries, which will be reported per manuscript sub-section below:

INTRODUCTION

1) Well arranged and sufficient information to guide concise summary of the disease issue and rationale for agent selection. 

2) Second paragraph, first sentence: A reference should be included for source of COVID-19 case numbers. 

3) Fourth paragraph, first sentence: It would be beneficial to note this is now focusing on data from 2003...on the initial reading, this reviewer missed you switched to discussing the first SARS outbreak and historical data. 

4) Fourth paragraph, "As in other virla diseases,...: : This sentence is phases oddly and would recommend re-writing for clarity. 

5) Fifth paragraph, "Mineralocorticoid receptor (MR) pathway..." : please include a reference for MR in multiple diseases versus just pulmonary or cardiac focus. 

6) Fifth paragraph, last sentence: Could be improved as not to end a sentence with a preposition. "...carried out in and limited data exists within real-world patient populations."

DISCUSSION

1) Concise summary of the results and application given the overall negative result.  

2) While intervention may trend with an improved/lower percentage of 6MWD, it can be commented how the difference in trial design/powering or adjusted doses would be clinically insignificant and/or not feasible for the doses necessary to observe an actionable change. 

Author Response

Dear Reviewer,

The authors would like to thank you for your valuable assessment of our manuscript. We have responded to each query from your review in a point-by-point letter. In the revised manuscript, we have included the exact wording of each query and our response. We thank you for your helpful comments and believe your suggestions have made the manuscript stronger.

Thank you for allowing me to review this manuscript and your clinical trial results evaluating the use of potassium canrenoate in the treatment/prevention of pneumonia and pulmonary fibrosis. 

Overall, while the trial has a general negative result, the presentation of the data is sound and appropriate conclusions have been drawn. It is in this reviewer's opinion that negative results in clinical trials are still results, and more trials of this type should be published. There is only a few minor commentaries, which will be reported per manuscript sub-section below:

INTRODUCTION

  • Well arranged and sufficient information to guide concise summary of the disease issue and rationale for agent selection. 

Authors’ Response: Thank you.

  • Second paragraph, first sentence: A reference should be included for source of COVID-19 case numbers. 

Authors’ Response: This has been improved and reference now [6] has been added: World Health Organization. WHO Coronavirus (COVID-19) Dashboard. [State on 6th January 2022] Available at: https://covid19.who.int

  • Fourth paragraph, first sentence: It would be beneficial to note this is now focusing on data from 2003...on the initial reading, this reviewer missed you switched to discussing the first SARS outbreak and historical data. 

Authors’ Response: Thank you for this interesting remark. We have added this information into the manuscript.

  • Fourth paragraph, "As in other virla diseases,...: : This sentence is phases oddly and would recommend re-writing for clarity. 

Authors’ Response: Thank you for this interesting remark. We modified the sentence to: “No specific cellular or molecular mechanisms leading to the development of pulmonary fibrosis in the course of SARS-CoV-2 infection, as in other viral diseases, have been identified.”

  • Fifth paragraph, "Mineralocorticoid receptor (MR) pathway..." : please include a reference for MR in multiple diseases versus just pulmonary or cardiac focus. 

Authors’ Response: Thank you for this valuable remark. We have added this information into the manuscript: “Mineralocorticoid receptor (MR) pathway activation contributes to the pathophysiology of many diseases, being able to activate specific intracellular genomic and nongenomic pathways, leading to regulation of homeostasis of the cardiovascular system and becoming a crucial regulator of the growth and function of different cell types (fibroblasts, cardiomyocytes, vascular cells)”. We added a reference here.

  • Fifth paragraph, last sentence: Could be improved as not to end a sentence with a preposition. "...carried out in and limited data exists within real-world patient populations."

Authors’ Response: Thank you for this valuable remark. We re-phrased this sentence and added information requested by another reviewer: The limitations of some of these studies are due to the use of animal models (rats or other rodents) or proposals derived from molecular and plant studies as new substances are being investigated.

DISCUSSION

  • Concise summary of the results and application given the overall negative result.  

Authors’ Response: Thank you for this valuable remark. The conclusions have been improved and now read as: “This RCT has shown that the administration of potassium canrenoate to patients with COVID-19 induced pneumonia was not associated with shorter mechanical ventilation time, shorter passive oxygenation, shorter length of hospitalization or less fibrotic changes on CT imaging. The overall mortality rate was not significantly different between the two groups. Adverse events recorded in this study were not significantly increased by the administration of potassium canrenoate The negative outcome of the study may be associated with a relatively small number of patients included. Any possible benefits from the use of potassium canrenoate as an antifibrotic drug in COVID-19 patients require further investigation.”

2) While intervention may trend with an improved/lower percentage of 6MWD, it can be commented how the difference in trial design/powering or adjusted doses would be clinically insignificant and/or not feasible for the doses necessary to observe an actionable change. 

Authors’ Response: Thank you for this very valuable remark. We included this in the limitations section of the manuscript: “Moreover, as the intervention may trend with an improved 6MWD, it may be argued that the difference in trial design achieved by including more patients, powering the study for improvement in 6MWD or adjusting the doses of the investigational drug would have shown clinical and statistical significance. “

With best regards

Reviewer 2 Report

Interesting study that reports negative results of a randomized clincial trial comparing potassium canrenoate versu placebo in COVID-19 inpatient. I suggest:

1) In the introduction the authors must increase the interest of spironolactone in COVID-19 by citing several studies (Kumar et al, viruses, 2021, nov 3, 13 (11):2209:Edwards et al, Front Endo, 2021, Nov 18, 12:747744); Perhaps a table synthezing the different studies (In Silico, In Vitro, Clinical) on spironolactone and potassium canrenoate could be useful.

2) In the introduction the authors could mention the repositionning of other FAD-approved drugs nammed FIASMAs (Loas & Le Corre, Pharmaceuticals, 2021) as one, amitriptyline has been found interesting in in vitro and clinical studies on pulmonary fibrosis (Zaafan et al, Naun Schmied Arch Plharmaco (PMID:30474696); Adams et al, Cellul physiol and Biochemis, 2016, 39, 565) and tested with success in human cells infected by the SARS-CoV-2 (see Carpinteiro et al, 2020, Cell Rep med, nov 17; 1 (!):100142.

3) It is not clear what is the principal criterion that lead to the calculation of the necessary number of subjects.  In statistcial analysis the authors must explain in details the calculation.

4) Why there is not equivalence of comorbidity taken into account before randomization? The difference of prevalence of Ischemia and intake of antidiabetic drugs could be a bias?

Author Response

Dear Reviewer,

The authors would like to thank the reviewer for their valuable assessment of our manuscript. We have responded to each query from your review in a point-by-point letter. In the revised manuscript, we have included the exact wording of each query and our response. We thank you for your helpful comments and believe your suggestions have made the manuscript stronger.

Interesting study that reports negative results of a randomized clincial trial comparing potassium canrenoate versu placebo in COVID-19 inpatient. I suggest:

  • In the introduction the authors must increase the interest of spironolactone in COVID-19 by citing several studies (Kumar et al, viruses, 2021, nov 3, 13 (11):2209:Edwards et al, Front Endo, 2021, Nov 18, 12:747744); Perhaps a table synthezing the different studies (In Silico, In Vitro, Clinical) on spironolactone and potassium canrenoate could be useful.

Authors’ Response: Thank you for this remark. We have added the citations, however, after trying to provide a table summarizing in vivo and vitro studies and after reading other reviews tending to shorten the introduction part, the authors agreed this table would be better suited for a review article or a commentary and is somewhat beyond the scope of an original article showing data from an RCT.

  • In the introduction the authors could mention the repositionning of other FAD-approved drugs nammed FIASMAs (Loas & Le Corre, Pharmaceuticals, 2021) as one, amitriptyline has been found interesting in in vitro and clinical studies on pulmonary fibrosis (Zaafan et al, Naun Schmied Arch Plharmaco (PMID:30474696); Adams et al, Cellul physiol and Biochemis, 2016, 39, 565) and tested with success in human cells infected by the SARS-CoV-2 (see Carpinteiro et al, 2020, Cell Rep med, nov 17; 1 (!):100142.

Authors’ Response: Thank you for this interesting remark. We have added these citations into the manuscript.

  • It is not clear what is the principal criterion that lead to the calculation of the necessary number of subjects.  In statistcial analysis the authors must explain in details the calculation.

Authors’ Response: Thank you for the important remark. It was not clearly phrased to reflect the aim of the study, so we rephrased the Statistical analysis section to adjust it accordingly. The full analysis included different options regarding the length of intubation and mechanical ventilation (please see table below, but the authors believe that adding this table to the manuscript is beyond the scope of this article). Now the section reads as follows:

Statistical analysis: “The sample size was calculated to demonstrate statistical significance of differences in the assessment of the duration of invasive mechanical ventilation via endotracheal intubation or tracheotomy (hours) at 48 hours after admission, assuming the standard significance level of the test p = 0.05 and power of 0.90. Additionally, it was assumed that the standard deviation (SD) of the length of the duration of invasive mechanical ventilation via endotracheal intubation or tracheotomy time would be 48 hours, and that when assessing statistical significance, the Student's t-test for independent samples was used. With the above information taken into the assessment by the statistician the study size was calculated to include 23 patients per arm, with a minimum of a total of 46 patients. The research project assumed the number of patients in each group should be at the level of 25, because this number was found to be achievable with the incurred costs, study time and availability of patients with predetermined inclusion and exclusion criteria. The participants were randomly divided into one of the two groups (according to the randomization table generated from the www.randomiser.com).”

Duration of invasive mechanical ventilation via endotracheal intubation or tracheotomy

Variant I (24-hour difference)

Variant I (48-hour difference)

Variant III (72-hour difference)

Mean  Mi1

120.00

120.00

120.00

Mean  Mi2

144.00

168.00

192.00

Standard deviation

48.00

48.00

48.00

Probability of alfa error

0.05

0.05

0.05

Power for the required N

0.90

0.91

0.92

Required number of patients (per group)

86

23

11

  • Why there is not equivalence of comorbidity taken into account before randomization? The difference of prevalence of Ischemia and intake of antidiabetic drugs could be a bias?

Authors’ Response: Thank you for this comment. We made every possible effort to match the patients in both groups to avoid bias, but this was not always possible on admission, during screening or inclusion into the study. Some additional information regarding co-morbidities was usually provided by the families or obtained from the GP during hospitalization. As ischemic heart disease status or the use of oral hypoglycemic drugs was not one of the exclusions criteria, we decided to include data regarding all recruited patients who finished the study. Moreover, although we report a difference between the use of oral hypoglycemic drugs the number of patients with diabetes was not statistically significant in both groups.

With best regards

Reviewer 3 Report

  1. The authors would add more to the COVID-19 different symptoms and complications.
  2. “the scientists around the world have focused on finding appropriate treatment of this specific type of pneumonia and its multiple long-term consequences” explain these consequences.
  3. For statistics from the World Health Organization (WHO) or any other source, please cite all relevant references.
  4. Simple explanation for the flowchart.
  1. Conclusion should be rewritten.
  2. The title should be informative and concise.
  3. In keywords: add “potassium canrenoate”
  1.  “COronaVIrus Disease” Chang ro “Corona Virus Disease”
  2. What does “ARDS” refer to?
  1. The authors could benefit from the following references;

Khalifa, S.A., Yosri, N., El-Mallah, M.F., Guo, R.G.Z., Musharraf, S.G., Du, M., Khatib, A., Xiao, J., Saeed, A., El-Seedi, H.H. and Zhao, C., 2020. Screening for natural and derived bio-active compounds in preclinical and clinical studies: one of the frontlines of fighting the coronaviruses pandemic. Phytomedicine, p.153311.

Author Response

Dear Reviewer,

The authors would like to thank the reviewer for their valuable assessment of our manuscript. We have responded to each query from your review in a point-by-point letter. In the revised manuscript, we have included the exact wording of each query and our response. We thank you for your helpful comments and believe your suggestions have made the manuscript stronger.

  1. The authors would add more to the COVID-19 different symptoms and complications.

Authors’ Response: Thank you for this remark. We have added more detailed symptoms and complications: “Majority of the cases are mild, viral illnesses. Yet, many patients suffer from severe respiratory diseases, i.e., pneumonia, respiratory failure, acute respiratory distress syndrome (ARDS) and non-respiratory symptoms, i.e. thrombosis and embolism, neuropathic pain, myositis, delirium, gastro-intestinal symptoms or long-term cognitive disturbances”.

  1. “the scientists around the world have focused on finding appropriate treatment of this specific type of pneumonia and its multiple long-term consequences” explain these consequences.

Authors’ Response: Thank you for this remark. We have explained the consequences: “Initially, it was difficult to identify and study the long-term consequences of SARS-CoV-2 infection, but many, including the most common - fatigue, muscle weakness, sleep difficulties, anxiety, depression and pulmonary fibrosis - are now becoming apparent.”

  1. For statistics from the World Health Organization (WHO) or any other source, please cite all relevant references.

Authors’ Response: Thank you for this remark. We have added the relevant reference. Authors’ Response: This has been improved and reference now [6] has been added: World Health Organization. WHO Coronavirus (COVID-19) Dashboard. [State on 6th January 2022] Available at: https://covid19.who.int

  1. Simple explanation for the flowchart.

Authors’ Response: Thank you for this remark. We have added more detailed information regarding the flowchart.

  1. Conclusion should be rewritten.

Authors’ Response: Thank you for pointing this out. We have altered the conclusions. This RCT has shown that the administration of potassium canrenoate to patients with COVID-19 induced pneumonia was not associated with shorter mechanical ventilation time, shorter passive oxygenation, shorter length of hospitalization or less fibrotic changes on CT imaging. The overall mortality rate was not significantly different between the two groups. Adverse events recorded in this study were not significantly increased by the administration of potassium canrenoate The negative outcome of the study may be associated with a relatively small number of patients included. Any possible benefits from the use of potassium canrenoate as an antifibrotic drug in COVID-19 patients require further investigation.”

  1. The title should be informative and concise.

Authors’ Response: Thank you for this remark. We have rephrased the title underline the negative results of the study. The title reads: “Mineralocorticoid Receptor Antagonist (Potassium Canrenoate) does not influence outcome in the treatment of COVID-19-associated pneumonia and fibrosis – a randomized placebo controlled clinical trial”.

  1. In keywords: add “potassium canrenoate”

Authors’ Response: Thank you. We have added potassium canrenoate into the keywords.

  1. “COronaVIrus Disease” Chang ro “Corona Virus Disease”

Authors’ Response: Thank you for this remark. We have changed the wording.

  1. What does “ARDS” refer to?

Authors’ Response: ARDS stands for Acute Respiratory Distress Syndrome. The abbreviation has been explained.

  1. The authors could benefit from the following references;

Khalifa, S.A., Yosri, N., El-Mallah, M.F., Guo, R.G.Z., Musharraf, S.G., Du, M., Khatib, A., Xiao, J., Saeed, A., El-Seedi, H.H. and Zhao, C., 2020. Screening for natural and derived bio-active compounds in preclinical and clinical studies: one of the frontlines of fighting the coronaviruses pandemic. Phytomedicine, p.153311.

Authors’ Response: Thank you for this interesting remark. We have added these citations and incorporated additional information into the manuscript.

With best regards

Reviewer 4 Report

Dear Authors, Although the article brings "negative" results, it is very useful to direct the treatment and research on other lines. The article seems well written. I would have some suggestions only for the introduction  -please improve the introduction, is  quite long and it should be more centered on the main subject - treatment - not that much on epidemiology and so on

Author Response

Dear Reviewer,

The authors would like to thank you for your thorough and valuable assessment of our manuscript. We believe your suggestions have made the manuscript stronger.

Reviewer's remark: Dear Authors, Although the article brings "negative" results, it is very useful to direct the treatment and research on other lines. The article seems well written. I would have some suggestions only for the introduction 

-please improve the introduction, is quite long and it should be more centered on the main subject - treatment - not that much on epidemiology and so on.

Authors’ Response: Thank you for your review and this valuable comment. We have improved the introduction to concentrate on the treatment, but we were also advised by other reviewers to add relevant studies, so the introduction is a bit lengthy. We still think all relevant information has been incorporated into the manuscript.

Round 2

Reviewer 2 Report

No additional changes are needed. Thus, the revised version of the manuscript can now be accepted.